# Superior reproducibility and repeatability in automated quantitative pupillometry compared to standard manual assessment, and quantitative pupillary response parameters present high reliability in critically ill cardiac patients

Benjamin Nyholm[1]*, Laust Obling[1], Christian Hassager[1], Johannes Grand[1], Jacob Møller[1], Marwan Othman[2], Daniel Kondziella[2,3], Jesper Kjaergaard[1]

1 Department of Cardiology, Rigshospitalet, Copenhagen University Hospital, Copenhagen, Denmark,
2 Department of Neurology, Rigshospitalet, Copenhagen University Hospital, Copenhagen, Denmark,
3 Faculty of Health and Medical Sciences, Department of Clinical Medicine, University of Copenhagen, Copenhagen, Denmark

* benjamin.nyholm@regionh.dk

## Abstract

### Background

Quantitative pupillometry is part of multimodal neuroprognostication of comatose patients after out-of-hospital cardiac arrest (OHCA). However, the reproducibility, repeatability, and reliability of quantitative pupillometry in this setting have not been investigated.

### Methods

In a prospective blinded validation study, we compared manual and quantitative measurements of pupil size. Observer and device variability for all available parameters are expressed as mean difference (bias), limits of agreement (LoA), and reliability expressed as intraclass correlation coefficients (ICC) with a 95% confidence interval.

### Results

Fifty-six unique quadrupled sets of measurement derived from 14 sedated and comatose patients (mean age 70±12 years) were included.

For manually measured pupil size, inter-observer bias was -0.14±0.44 mm, LoA of -1.00 to 0.71 mm, and ICC at 0.92 (0.86–0.95). For quantitative pupillometry, we found bias at 0.03±0.17 mm, LoA of -0.31 to 0.36 mm and ICCs at 0.99. Quantitative pupillometry also yielded lower bias and LoA and higher ICC for intra-observer and inter-device measurements.

Correlation between manual and automated pupillometry was better in larger pupils, and quantitative pupillometry had less variability and higher ICC, when assessing small pupils.

**Data Availability Statement:** The data collection was performed as part of the quality assessment for clinical procedures and equipment, approved by the hospital administration. Formal informed consent is therefore waived as per Danish legislation. Hence, data cannot be shared publicly according to Danish legislation. To gain data access, a formal request can be sent to the contact person of the Clinical Research Units for cardiovascular research at the Department of Cardiology, Copenhagen University Hospital Rigshospitalet. The institute will obtain an individual permission from the Danish National Ethics Committee for researchers who meet the criteria for access to confidential patient data. Contact persons for the Clinical Research Units: Ginette Wedel, Secretary Phone: +45 35 45 26 64 Email: ginette.wedel@regionh.dk.

**Funding:** BN and JK are supported by a grant from the "Novo Nordisk Foundation" (Grant no. NNF17OC0028706), "https://novonordiskfonden. dk". The funders had no role in study design, data collection and analysis, decision to publish, or preparation of the manuscript.

**Competing interests:** The authors have declared that no competing interests exist.

Further, observers failed to detect 26% of the quantitatively estimated abnormal reactivity with manual assessment.

We found ICC >0.91 for all quantitative pupillary response parameters (except for latency with ICC 0.81–0.91).

## Conclusion

Automated quantitative pupillometry has excellent reliability and twice the reproducibility and repeatability than manual pupillometry. This study further presents novel estimates of variability for all quantitative pupillary response parameters with excellent reliability.

## Introduction

Critically ill patients admitted to the cardiac intensive care unit (cICU) commonly experience neurological complications [1,2]. Especially in patients resuscitated from out-of-hospital cardiac arrest (OHCA) anoxic brain injury is the leading cause of death [3,4].

It is challenging to identify these patients with poor neurological outcome early [5,6], and evaluation of pupil size and reactivity is of great prognostic importance [7,8]. When additional neurological evaluation (imaging of the brain, electroencephalography and somatosensory evoked potential) is needed before deciding about withdrawal of life-sustaining therapy, the timing can be guided by results from serial pupillometry as part of accurate multidisciplinary neuroprognostication [7–9]. Hence, great reliability for pupillometry is essential.

In the clinical setting, pupillary assessments are often performed manually using a penlight for reactivity and a pupil gauge for pupil size, even though several studies investigating reliability find that automated quantitative pupillometry is superior to manual pupillometry [10–17]. In current American and European resuscitation guidelines [5,6], quantitative pupillometry is recommended as part of the multimodal prognostication of comatose patients resuscitated from OHCA. However, these guidelines recognize that evidence is still limited, and the lack of knowledge on variability of quantitative pupillometry may inflict on its clinical usefulness and the subsequent decision-making [18]. Furthermore, most studies predicting outcome of morbidity and mortality with quantitative pupillometry rely only on the overall pupillary reactivity (degree of contraction) [15,17]. Hence, reliability for all the individual quantitative pupillary response parameters (including contraction and dilation velocity etc.) have not been thoroughly investigated [10–16]. However, a recent post-hoc analysis from a multicenter prospective observational study found that several individual parameters were associated with poor neurologic outcome [19].

To accommodate the demand for knowledge of variability, we investigated the observer and device reproducibility, repeatability and reliability of quantitative pupillometry in a clinical setting at a specialized cICU and compared the performance to the standard manual assessment. Additionally, we aimed to present novel estimates of variability for all the individual quantitative pupillary response parameters of an automated pupillometer for future scientific and clinical use.

## Methods

### Study design and population

We conducted a single-center prospective double-blinded validation study in a cICU at a tertiary heart center. Throughout August and September 2020, all patients admitted to the cICU

were considered eligible for inclusion and otherwise treated in agreement with guidelines [5]. This comprised hemodynamically unstable patients requiring specialized intensive cardiac care. No elective patients were included.

No patients fell for exclusion, with any preexisting ophthalmic condition that would hinder examination to both eyes were found.

The aim was to include a minimum of 20 quadruple measurements to cover the range of pupil sizes, a pragmatic sample size used in previous validation studies [20,21].

## Standard manual and quantitative pupillometry

Manual pupillary assessments were performed using a penlight for reactivity and a pupil gauge for measuring pupil size. Pupil size was recorded on an ordinal scale from 1 to 10 millimeters, and the pupil reactivity was graded as an absent, sluggish, normal and hyperreactive response. No hyperreactive response were registered and this category was omitted from further analysis.

Quantitative pupillometry measurements were performed using two NPi®-200 pupillometers (NeurOptics®, Irvine, CA, USA). With a calibrated light stimulation of fixed intensity (1000 Lux) and duration (3.2 s), the device uses an infrared camera to achieve a rapid measure (0.05 mm limit) of the pupil size and a series of several dynamic pupillary variables. A single-use plastic 'chin guard' (SmartGuard®) was used for every pupillary measurement, ensuring a uniform distance from the eye and the pupil targeted on an LCD screen. The quantitative assessment of the pupillary light reflex (PLR) divides the reactivity into parameters of latency of constriction (LAT, seconds), percental change (%CH, %), average/maximum constriction velocity (CV/MCV, millimeters/seconds), and the subsequent dilation velocity (DV, millimeters/seconds), besides maximum/absolute and minimum pupil size (MAX/MIN, millimeters) [22]. As previously, abnormal pupil reactivity is defined as %CH <15% [14].

Reactivity can be expressed as Neurological Pupil index[TM] (NPi), a scalar value (between 0 and 5) derived from an algorithm including quantitative parameters [23]. The higher the NPi score, the more reactive response, and on the contrary the lower the NPi score, the more abnormal response [23]. For the NPi value, a score <3 is considered abnormal (a sluggish response) and a score ≥3 is considered normal (a brisk response).

All parameters were measured automatically and collected in the SmartGuard® data storage device. The pupillometer measures human pupils sizing varying from 1 mm to 9 mm, with an accuracy of 0.03 mm [22].

## Terminology of measurement error

In this study we use different terms describing studies of measurement error.

When measurements are made on the same subject *agreement* is the quantification of how close measurements are to each other, with the error given in the same scale as the measurements. It is independent of the population and refers to the methods used. *Reliability*, however, relates the observed measurement errors to the, "true", inherent variability between subjects, and depends on population heterogeneity. In this study we evaluate reliability with intraclass correlation coefficient (ICC) [20,24,25].

*Repeatability* refers to the variability in repeated measurements to the same subject under the same conditions, and *reproducibility* to the variation when conditions are different between measurements. Contentions differ in relation to the desired point of interest as in this study when investigating measurements between different observers and devices [26].

### Patient assessment procedure

To collect pupillometry in the clinical setting patients at the cICU were selected for additional study-related pupillary measurements in the period from admission through the initiation and discontinuation of sedation, targeted temperature management (TTM) to 36 degrees Celsius, and treatment with vasoactive agents, to discharge from the cICU or withdrawal of life-sustaining treatment. Patients were subjected to multiple assessments during their admission.

During the inclusion period, pairs of two trained nurses (observers) from the cICU staff pool (n = 24),performed sets of quadruple measurement-pairs, each consisting of a manual and then a quantitative pupillometry measurement to both left and right eye, in a pre-specified sequence, all completed within 5 minutes. The *observer reproducibility* (inter-observer variability) was addressed by the two nurses, each performing measurements (measurement-pairs #1 and #2) subsequently to each other. The *observer repeatability* (intra-observer variability) and *device repeatability* (intra-device variability, only for quantitative pupillometry) were investigated by comparing the first measurements from the first nurse with the repeated measures from the same nurse (measurement-pairs #1 and #3). Finally, *device reproducibility* (inter-device variability, only for quantitative pupillometry) was addressed by comparing the first measurements from the second nurse with repeated measurement of quantitative pupillometry, but with a second pupillometer (measurement-pairs #2 and #4). The sequence of each patient's set of quadruple measurement-pairs is presented in Fig 1.

Each observer used REDCap (Research Electronic Data Capture) [27,28] surveys to enter the manual measurements directly in a database immediately after an assessment, blinding the data to the other observers. Results from quantitative pupillometry were automatically stored directly in the pupillometers SmartGuard, blinded for all observers. All data were anonymized and blinded for the outcome assessors analyzing the data.

The data collection was performed as part of the quality assessment for clinical procedures and equipment, approved by the hospital administration. Formal informed consent is therefore waived as per Danish legislation. The study conforms to the guiding principles of the Declaration of Helsinki and all authors have read and approved the manuscript in its present

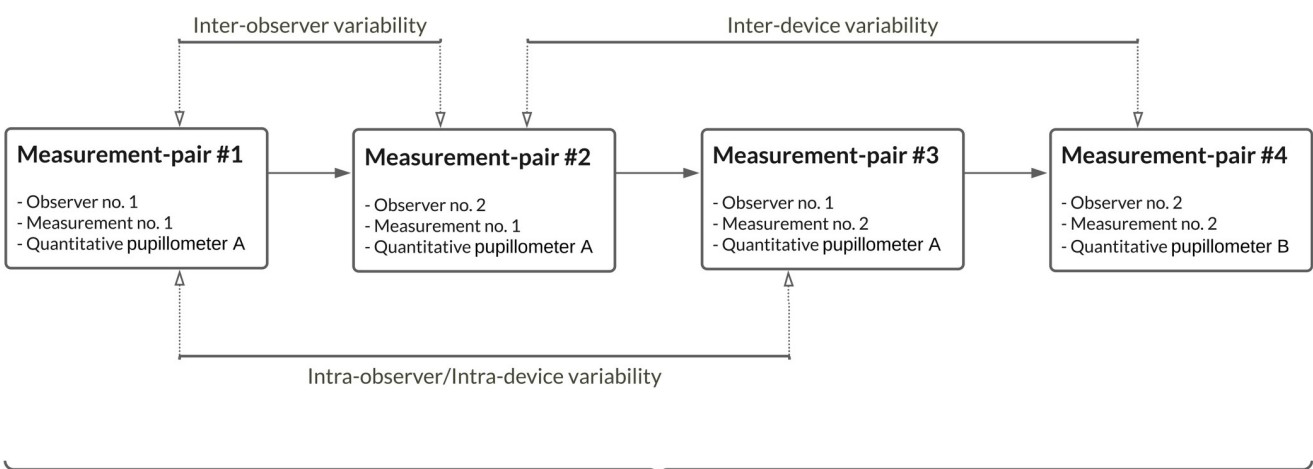

**Fig 1. Patient assessment procedure.** Flowchart of "Patient assessment procedure" depicting a complete set of quadruple measurement-pairs to the same patient within 5 minutes, with every measurement-pair comprising one manual, and one quantitative measurement, to both left and right eye, subsequently, blinded to the other observer.

form. The manuscript is not under consideration for publication elsewhere and none of the results have previously been published. All authors meet the authorship criteria.

## Statistical analysis

Continuous variables were tested for normality by visualization of the histogram presentation of data and presented as mean values ± standard deviation (SD). Categorical variables are presented as frequency and percentage.

Absolute variation between measurements were analyzed with Bland-Altman (BA) statistics and presented in plots including mean differences (bias) adapted 95% limits of agreement (LoA), as ±1.96 SD [20]. For test of reliability we use ICC presented with 95% confidence interval (ICC (95% CI)) [29]. ICC values less than 0.50 were interpreted as poor, values between 0.50 and 0.75 as moderate, values between 0.75 and 0.90 as good, and values greater than 0.90 as excellent reliability [30].

As sensitivity analysis, we stratified pupil sizes into two groups of "small" and "large" pupils (below and above median pupil size of 2.2 mm) and into left and right eye and repeated the analyses.

When investigating agreement between measurements of the two methods we used BA statistics and tested the association between measurements of categorical and continuous variables with Spearman's rank correlation test.

All statistical analyses were performed using R version 4.0.3 [31], and a p-value <0.05 was considered significant.

## Results

### Baseline characteristics

We included 14 patients, either sedated or comatose, with a mean age of 70±12 years. Patients were predominantly males (57%), 10 (71%) were resuscitated from OHCA, and 4 were other critically ill patients admitted to the cICU. We collected 56 sets of quadruple measurement-pairs for analysis, wherein 14 (25%) pairs were obtained while the patients underwent TTM to 36 degrees Celsius, 44 (79%) while patients were sedated, and 46 (82%) while patients received a vasoactive agent. All baseline characteristics are presented in Table 1.

### Standard manual versus quantitative pupillometry—Pupil size

Intra-, and inter-observer variability for maximum size using standard manual pupillometry had a bias of -0.02±0.31 mm and LoA of -0.58 to 0.63 mm, and bias at -0.14±0.44 mm with LoA of -1.00 to 0.71 mm, respectively.

Intra-observer variability for maximum pupil size measured with quantitative pupillometry was associated with a bias of 0.04±0.13 mm with LoA of -0.21 to 0.28 mm, and for minimum pupil size, a bias of 0.01±0.13 mm with LoA of -0.24 to 0.26 mm. Assessment of inter-observer variability resulted in a bias of 0.03±0.17 mm with LoA of -0.31 to 0.36 mm, and a bias of 0.03 ±0.18 mm with LoA of -0.33 to 0.38 mm for maximum, and minimum pupil size, respectively.

Intra-, and inter-observer measurements of manual measurements resulted in ICC at 0.89 (0.83–0.93), and 0.92 (0.86–0.95), respectively. Quantitative pupillometry produced ICCs at 0.99 for all measurements. All individual results of pupil size for manual and quantitative pupillometry are presented in Table 2 with BA plots in Fig 2.

When comparing manual and quantitative assessment of pupil size, we found bias at -0.02 ±0.45 mm with LoA of -0.91 to 0.87 mm, and ICC at 0.90 (0.87–0.92). We found significant correlation between manual and quantitative pupillometry for both intra-observer

**Table 1. Baseline characteristics.**

| Patients | n = 14[a] |
|---|---|
| **Demography** | |
| Age (years) | 70±12 |
| Male sex | 8 (57%) |
| **Prior medical/surgical conditions** | |
| Arrhythmia | 3 (21%) |
| Chronic obstructive pulmonary disease | 3 (21%) |
| Diabetes | 2 (14%) |
| Ischemic heart disease | 5 (36%) |
| Heart failure | 3 (21%) |
| Hypertension | 8 (57%) |
| Valvular heart disease | 1 (7%) |
| **Admission diagnosis** | |
| Out-of-hospital cardiac arrest | 10 (71%) |
| Cardiogenic shock | 2 (14%) |
| ST elevation myocardial infarction | 1 (7%) |
| Epidural hematoma | 1 (7%) |
| **Procedures performed** | |
| Coronary angiography | 11 (79%) |
| Percutaneous coronary intervention | 7 (50%) |
| Left ventricular ejection fraction by echocardiography, % | 32±16 |
| **Patient assessments** | **n = 56a[a]** |
| **Treatment during measurement** | |
| Sedating agents | 44 (79%) |
| Propofol | 28 (50%) |
| Fentanyl | 30 (54%) |
| Remifentanil | 14 (25%) |
| Midazolam | 2 (4%) |
| Dexmedetomidin | 2 (4%) |
| Other opioids | 8 (14%) |
| Targeted temperature management | 14 (25%) |
| Vasoactive agents | 46 (82%) |
| Norepinephrine | 44 (79%) |
| Dopamine | 6 (11%) |
| Other | 14 (25%) |

[a]Statistics presented: Mean±SD, n (%).

SD = standard deviation.

(Spearman's rho = 0.63, 95%CI: 0.36–0.80; p<0.001) and inter-observer (Spearman's rho = 0.71, 95%CI: 0.49–0.86; p <0.001) measurements of pupil size.

For measurements stratified according to pupil size below and above 2.2 mm, quantitative pupillometry still presented higher ICCs, with lower values of LoA in both groups.

Comparing manual and quantitative assessment in the different size groups, we found correlation of methods for both small pupils (Spearman's rho = 0.65, 95%CI: 0.35–0.83; p<0.001 for intra-observer, and Spearman's rho = 0.53, 95%CI: 0.17–0.77; p = 0.003 for inter-observer), and for large pupils (Spearman's rho = 0.72, 95%CI: 0.26–0.94; p<0.010 for intra-observer, and Spearman's rho = 0.85, 95%CI: 0.45–0.97; p<0.001 for inter-observer).

**Table 2. Statistical analysis.**

| | Manuel Pupillometry | Quantitative Pupillometry | | | | | | | |
|---|---|---|---|---|---|---|---|---|---|
| | Maximum diameter, mm | Maximum diameter, mm | Minimum diameter, mm | Percental change, % | Constriction Velocity, mm/s | Maximum Constriction Velocity, mm/s | Dilation Velocity, mm/s | Latency of constriction, s | Neurological Pupil index |
| Bias, mean±SD | | | | | | | | | |
| Intra-observer | 0.02±0.31 | 0.04±0.13 | 0.01±0.13 | 0.84±3.27 | 0.03±0.25 | 0.07±0.30 | 0.01±0.10 | <0.01±0.04 | 0.04±0.21 |
| Inter-observer | -0.14±0.44 | 0.03±0.17 | 0.03±0.18 | -0.26±4.42 | -0.06±0.33 | -0.08±0.43 | <0.01±0.12 | <0.01±0.02 | -0.02±0.21 |
| Inter-device | - | 0.01±0.17 | -0.02±0.15 | 1.42±4.47 | 0.06±0.32 | 0.14±0.39 | 0.03±0.10 | <0.01±0.04 | 0.04±0.19 |
| Mean value, mean±SD | | | | | | | | | |
| Intra-observer | 2.11±0.54 | 2.17±0.73 | 1.74±0.43 | 18.29±9.91 | 1.17±0.87 | 1.66±1.25 | 0.41±0.31 | 0.24±0.04 | 4.42±0.40 |
| Inter-observer | 2.23±0.81 | 2.28±0.85 | 1.78±0.48 | 19.70±10.16 | 1.28±0.88 | 1.85±1.38 | 0.46±0.35 | 0.24±0.04 | 4.47±0.36 |
| Inter-device | - | 2.02±0.31 | 1.66±0.28 | 17.32±8.87 | 1.02±0.47 | 1.45±0.77 | 0.35±0.23 | 0.24±0.04 | 4.45±0.41 |
| Limit of Agreement (lower, upper) | | | | | | | | | |
| Intra-observer | -0.58, 0.63 | -0.21, 0.28 | -0.24, 0.26 | -5.56, 7.25 | -0.45, 0.52 | -0.52, 0.66 | -0.19, 0.21 | -0.07, 0.07 | -0.36, 0.45 |
| Inter-observer | -1.00, 0.71 | -0.31, 0.36 | -0.33, 0.38 | -8.93, 8.41 | -0.69, 0.58 | -0.93, 0.77 | -0.24, 0.24 | -0.05, 0.05 | -0.43, 0.40 |
| Inter-device | - | -0.33, 0.35 | -0.32, 0.27 | -7.35, 10.19 | -0.56, 0.69 | -0.63, 0.91 | -0.18, 0.23 | -0.07, 0.08 | -0.34, 0.42 |
| Intraclass Correlation Coefficients (95% CI) | | | | | | | | | |
| Intra-observer | 0.89 (0.83–0.93) | 0.99 (0.99–1.00) | 0.98 (0.97–0.99) | 0.97 (0.96–0.98) | 0.98 (0.97–0.99) | 0.99 (0.98–0.99) | 0.98 (0.96–0.98) | 0.81 (0.70–0.88) | 0.93 (0.88–0.95) |
| Inter-observer | 0.92 (0.86–0.95) | 0.99 (0.98–0.99) | 0.96 (0.94–0.98) | 0.95 (0.92–0.97) | 0.96 (0.94–0.98) | 0.97 (0.96–0.98) | 0.97 (0.95–0.98) | 0.92 (0.88–0.95) | 0.91 (0.86–0.94) |
| Inter-device | - | 0.99 (0.98–0.99) | 0.97 (0.95–0.98) | 0.95 (0.92–0.97) | 0.97 (0.95–0.98) | 0.98 (0.97–0.99) | 0.98 (0.97–0.99) | 0.82 (0.72–0.89) | 0.94 (0.90–0.96) |

SD = standard deviation, 95% CI = 95% confidence interval, mm = millimeters, s = seconds.

When addressing pupil size for left and right eyes isolated, manual pupillometry presented intra-observer variability with LoA at -0.59, 0.63 mm (for both left and right), and inter-observer variability with LoA at -1.10, 0.80 mm (left) and -0.95, 0.83 mm (right). Intra-observer variability for quantitative pupillometry resulted in a LoA at -0.19, 0.24 mm (left) and -0.23, 0.32 mm (right), and inter-observer variability with LoA at -0.33, 0.39 mm (left) and -0.30, 0.34 mm (right), respectively.

ICC for manual pupillometry was found at 0.89 (95%CI: 0.80–0.94) for intra-observer measurements in both eyes. For inter-observer measurements we found ICC at 0.91 (95%CI: 0.82–0.95) and 0.92 (95% CI: 0.84–0.96) for left and right eye respectively. Quantitative pupillometry yielded ICC >0.98 (95%CI: 0.98–0.99) for intra-, and inter-observer measures in both eyes.

## Standard manual versus quantitative pupillometry–Pupil reactivity

When measuring pupil reactivity with manual assessments, ICC was 0.89 (0.84–093) and 0.76 (0.63–0.85) for intra-, and inter-observer respectively. Reactivity measured with quantitative pupillometry (%CH) produced intra-, and inter-observer ICC at 0.97 (0.96–0.98) and 0.95 (0.92–0.97) respectively.

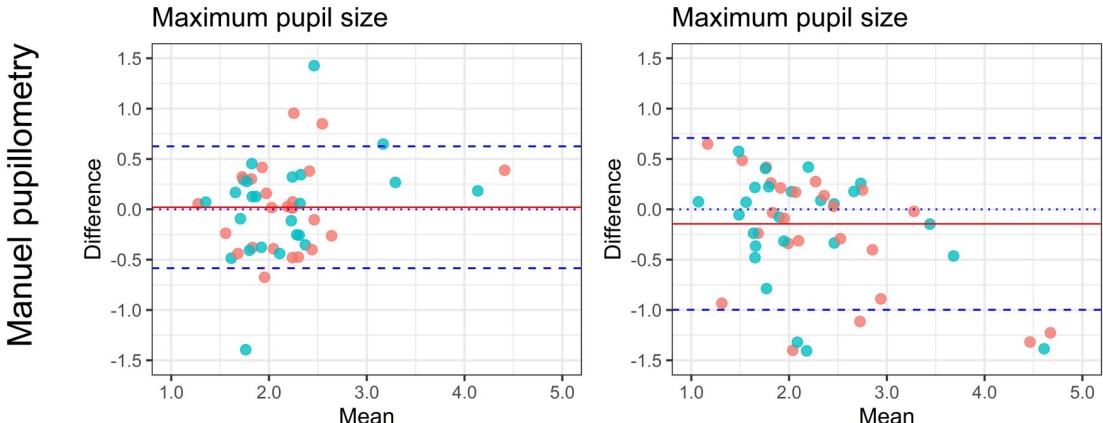

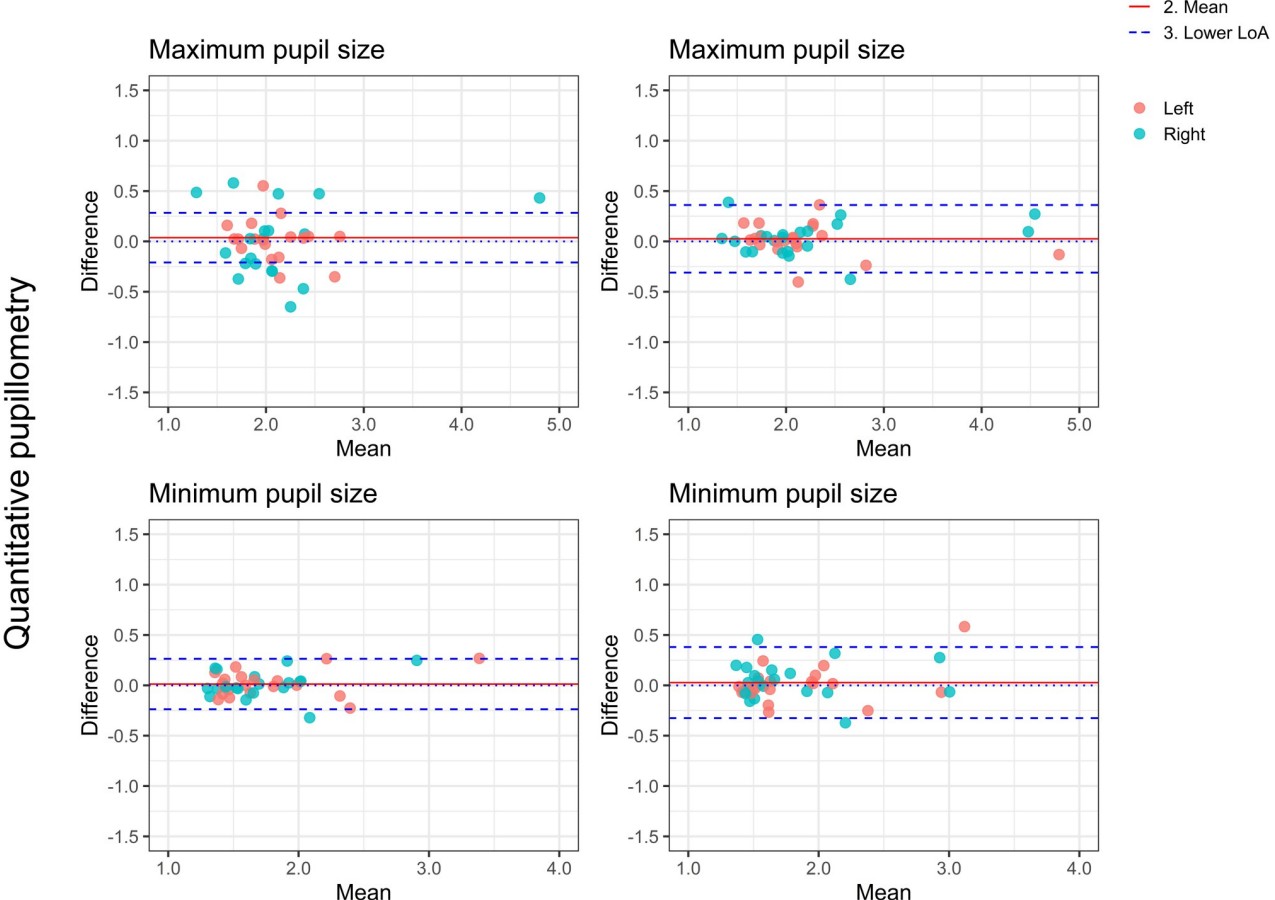

**Fig 2. Standard manual versus quantitative pupillometry—Pupil size.** Bland Altman plots depicting intra-, and inter-observer variability of pupil size measured with manual and quantitative pupillometry.

When comparing manual reactivity and %CH we found a correlation for both intra-observer (Spearman's rho at 0.66, 95%CI: 0.43–0.80; p <0.001) and for inter-observer measurements (Spearman's rho at 0.69, 95%CI: 0.47–0.84; p <0.001). In 26% of the cases when quantitative pupillometry detected abnormal pupil reactivity (%CH <15%), a normal reactivity was noted for manual pupillometry.

In measurements performed while patients were sedated the mean %CH value was 17% and 25% when not sedated, although lower with sedation, this difference was not significant (p = 0.234). When testing reliability for measurements without sedation, we found an ICC at 0.98 (0.95–0.99) and 0.99 (0.96–0.99) for intra-observer and inter-observer measurements, respectively. For the sedated we found ICC at 0.96 (0.94–0.98) and 0.92 (0.86–0.95) for intra-observer and inter-observer measurements.

### Additional quantitative pupillary response parameters

The statistical results of NPi and the additional quantitative pupillary response parameters are shown in Table 2 and presented with BA plots in Fig 3.

The NPi value presented a bias at -0.05±0.21 with LoA of -0.36 to 0.45 for intra-observer variability, and at -0.02±0.21 with LoA of -0.43 to 0.40 for inter-observer variability. NPi presented ICCs at 0.93 (0.88–0.95) and 0.91 (0.86–0.94) for intra-, and inter-observer

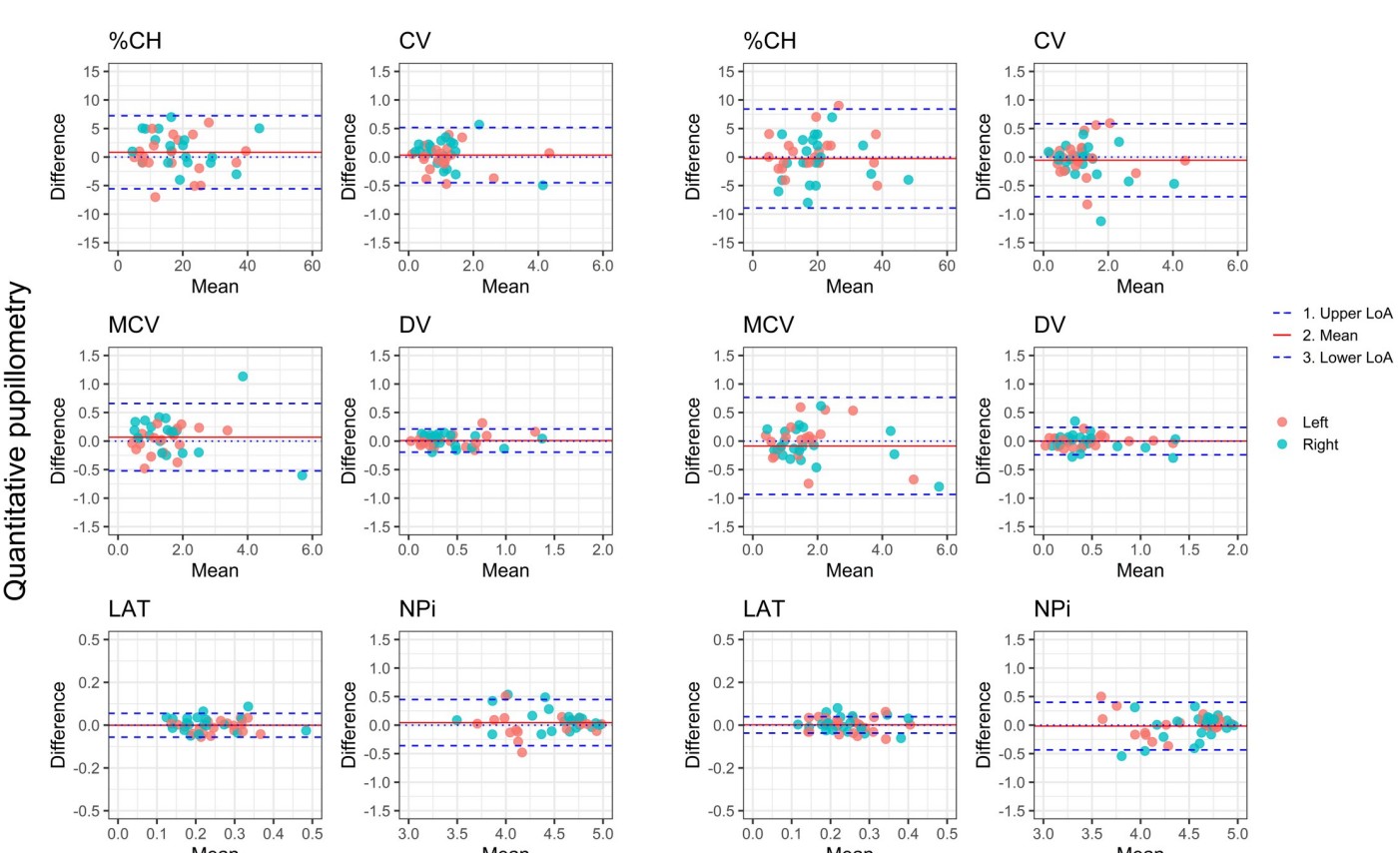

**Fig 3. Additional quantitative pupillary response parameters.** Bland Altman plots depicting intra-, and inter-observer variability of quantitative pupillary response parameter. %CH = percental change, CV = average constriction velocity, MCV = maximum constriction velocity, DV = dilation velocity, LAT = latency of constriction, NPi = Neurological Pupil index, LoA = limit of agreement.

measurements, respectively. When assessing the remaining quantitative parameters, we found low bias and LoA narrow, relative to the mean, and the ICCs for all parameters were good to excellent with values at 0.81 (0.70–0.88) and 0.92 (0.88–0.95) for LAT, and 0.95 to 0.99 for all other parameters.

### Device reproducibility, repeatability and reliability

The bias of maximum and minimum pupil size between the two quantitative pupillometry devices (inter-device variability) was measured to be 0.01±0.17 mm with LoA of -0.33 to 0.35 mm and -0.02±0.15 mm with LoA of -0.32 to 0.28 mm, respectively. The ICCs was at 0.99 (0.98–0.99).

The inter-device measurements of NPi resulted in a bias of -0.04±0.20 with LoA of -0.34 to 0.42, combined with an ICC at 0.94 (0.90–0.96). Overall reliability of all the quantitative pupillary response parameters for inter-device measurements resulted in ICC at 0.94–0.99 (except for LAT with ICC at 0.82 (0.72–0.89)).

Intra-device (intra-observer) measurements are presented in the previous chapter, and, collectively with inter-device measurements, are further shown in Table 2 with BA plots in Fig 4.

## Discussion

In our study of unconscious and critically ill patients, primarily with cardiac OHCA, we found twice the observer reproducibility and repeatability for quantitative measurements of pupil size with better measurements of reliability, for both size and reactivity, compared to the standard manual assessment.

When investigating the individual agreement (both intra-, and inter-observer) of the two methods, we found that assessment of the pupil size by the standard manual method by experienced observers, resulted in LoA twice as wide compared to the automated pupillometer. Furthermore, the ICC values suggest that intra- and inter-observer reliability was lower for the manual assessments than for the automated. These results corroborate with earlier data of greater variability in measurements of the standard manual method verifying the inferiority compared to automated quantitative pupillometry [10–15]. As a consequence of this variability Couret et al. [14] revealed that observers, using standard manual examinations, missed 50% of anisocoria (difference in pupil size between left and right eye, here defined as >1.0 mm) in patients and falsely detected 16 cases compared to the automated pupillometry. In concordance with this, we found that measurements of automated quantitative pupillometry have higher reproducibility and repeatability, than manual assessment, for both left and right eye isolated.

There is some disagreement regarding the correlation between standard manual and automated pupillometer in the literature. Meeker et al. and Couret et al. [11,14,16] report a poor correlation of the two methods in contrast to Yan et al and latest Smith et al. [32,33] that present a closer correlation. A fundamental problem with the manual pupillometry is that the reaction and difference in size is relatively smaller in smaller pupils, and thereby is more difficult to identify. In concordance, poor correlation was most pronounced when assessing small pupils for absolute size and reactivity (39% discordance for pupil size < 2.0 mm, and 4% in pupil size >4 mm in Couret et al.), and Yan et al. and Smith et al. did not stratify for size [14,32,33]. This is supported by our results of better correlation between the methods in larger pupils and measurements with higher reliability and less variability for the quantitative assessment than for manual, when assessing small pupils.

Smith et al. [33] find it unlikely that the small mean difference (bias 0.15 with LoA of ±1.4) in pupil size found with manual assessment in their study should result in a clinically

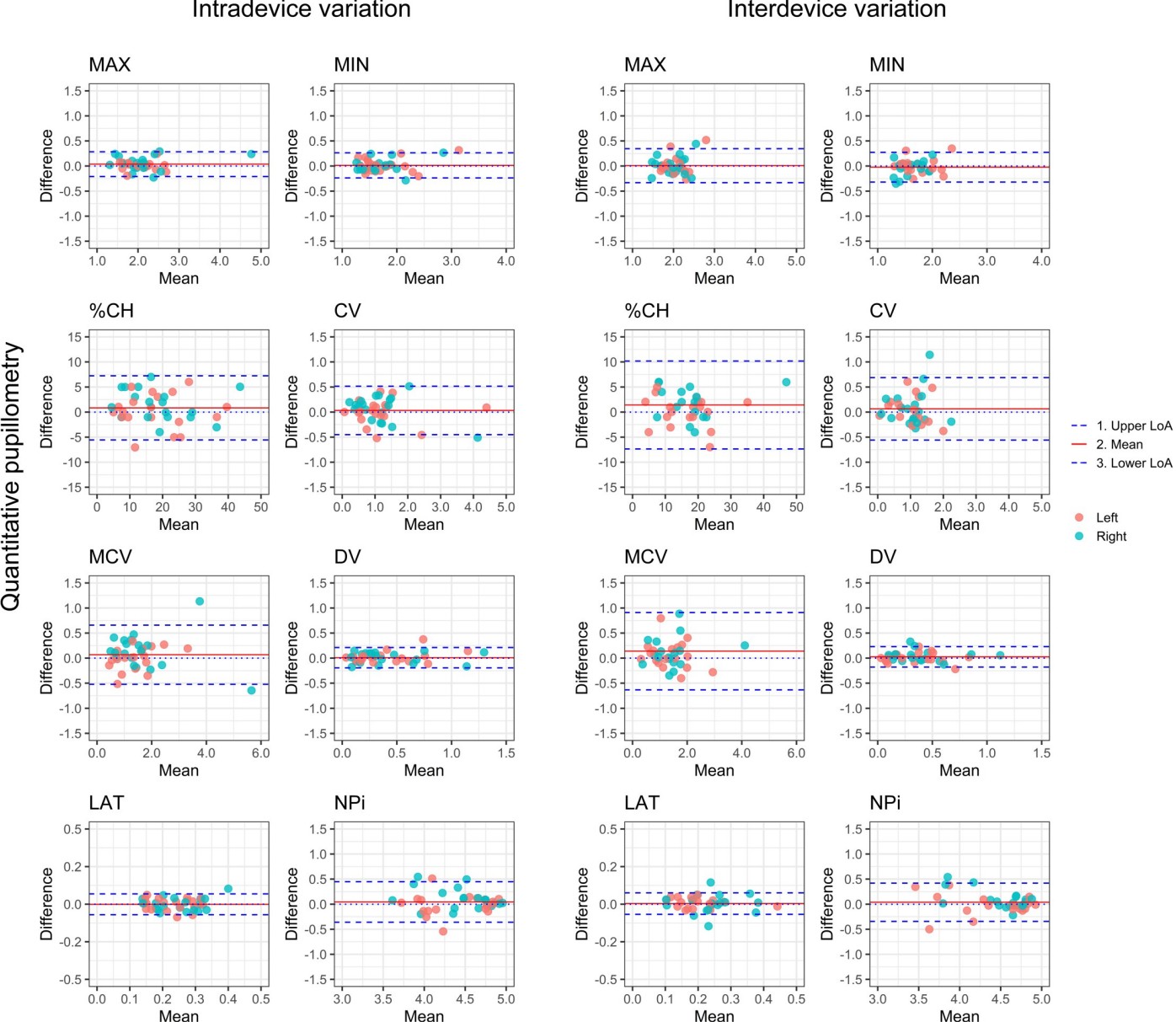

**Fig 4. Device reproducibility and repeatability.** Bland Altman plots depicting intra-, and inter-device variability of quantitative pupillometry. MAX size = maximum pupillary diameter, MIN size = minimum pupillary diameter, %CH = percental change, CV = average constriction velocity, MCV = maximum constriction velocity, DV = dilation velocity, LAT = latency of constriction, NPi = Neurological Pupil index, LoA = limit of agreement.

significant outcome. However, when the cutoff point used to score anisocoria is set to 1.0 mm (Couret et al.) or 0.5 mm (Taylor et al.), a variability (LoA) of ±1.4 mm (Couret et al.), or even ±0.9 mm as found in our study, could prove to be a significant margin of error in the clinical setting [14,34].

While automated quantitative pupil assessment outperforms the standard manual evaluation, even a LoA of ±0.3 mm, with a pupillometer accuracy of 0.03 mm [22], for a pupil with a mean size of 2.2 mm, contracting to a minimum size of 1.7 mm, should be taken in to account.

The quantitative pupillometer yielded excellent reliability for measurement of reactivity (%CH), even under sedation, compared to a considerably lower value for the manual

assessments, and when the two methods were compared, the reliability was only moderate. We found that observers failed to detect 26% of the quantitatively estimated abnormal PLR with the manual assessment, in concordance with an error rate at 18% for Couret et al. [14].

This lack of reliability and inaccuracies in estimating pupil size, and reactivity in manual pupil assessment with a penlight is problematic, as this still seems to be the standard regime for evaluating pupil size and reactivity in the clinical setting. If pupillometry is to be included in an accurate post-CA neuroprognostication protocol, it warrants the clinical application of automated quantitative pupillometry in favor of the standard manual evaluation.

A recent study concludes that measurements of pupil size, constriction, and latency were not always interchangeable in the two different devices studied [10]. This could be a critical challenge as several devices are often used interchangeably in the clinical setting and relied upon for multimodal prognostication. However, our study finds excellent inter-device reliability for all the individual quantitative pupillary response parameters, with overall identical results regarding size and reactivity as for intra- and inter-observer measurements. These results are consistent with the only other study prior to this [13].

For all the additional quantitative pupillary response parameters, recently presented with promising prognostic value by Tamura et al. [19], this study offers novel data on reproducibility and repeatability."

Overall, this study support that automated quantitative pupillometry performs well in the clinical setting even when several observers, multiple patients and separate pupillometry devices are involved. However, further studies, implementing the additional quantitative pupillary response parameters, are needed.

## Limitations

As stated, the 56 sets of quadruple pupil measurement-pairs were obtained from 14 patients; hence, measurements are in themselves not entirely independent, which may underestimate the inherent variability between subjects. The prevalence of extreme pupil sizes (i.e., very small or very large initial pupil size) was low; however, we still cover the majority of pupil sizes in the current sample.

The PLR can be affected by any interference of the balance of autonomic control of pupil size. Thus, any anatomical, physical or pharmacological condition that challenges this balance, can affect the pupillary size and reactivity pattern [18,35,36]. Our pupillary measurements were made by experienced observers in a clinical setting at the cICU, with or without sedation, TTM and vasoactive agents. In this setting, no control could be made regarding the type or amount of anesthesia or inotropic agents given, or the exact timing of pupillometry measurements. However, individual measurements for comparison were made at the same patients within 5 minutes, keeping similar conditions regarding the state of sedation, opioid treatment, and ambient lighting. Other studies found no difference in PLR agreement for patients with or without sedation when assessed within a short time frame [14,36].

## Conclusion

In this prospective blinded validation study, we found excellent reliability and twice the reproducibility and repeatability for automated quantitative pupillometry compared to manual pupillometry.

We present novel estimates of variability for all quantitative pupillary response parameters with excellent reliability.

## Author Contributions

**Conceptualization:** Benjamin Nyholm, Christian Hassager, Marwan Othman, Daniel Kondziella, Jesper Kjaergaard.

**Data curation:** Benjamin Nyholm, Laust Obling, Johannes Grand.

**Formal analysis:** Benjamin Nyholm.

**Funding acquisition:** Jesper Kjaergaard.

**Investigation:** Benjamin Nyholm, Laust Obling.

**Methodology:** Benjamin Nyholm, Jesper Kjaergaard.

**Project administration:** Benjamin Nyholm, Jesper Kjaergaard.

**Resources:** Benjamin Nyholm.

**Software:** Benjamin Nyholm.

**Supervision:** Christian Hassager, Jacob Møller, Daniel Kondziella, Jesper Kjaergaard.

**Validation:** Benjamin Nyholm, Laust Obling, Jesper Kjaergaard.

**Visualization:** Benjamin Nyholm.

**Writing – original draft:** Benjamin Nyholm, Laust Obling, Christian Hassager, Johannes Grand, Jacob Møller, Marwan Othman, Daniel Kondziella, Jesper Kjaergaard.

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
