## [Decision Letter · Decision Letter 0]

8 Feb 2022

PONE-D-22-00255Observer and device reproducibility, repeatability, and reliability of automated quantitative pupillometry in critically ill cardiac patientsPLOS ONE

Dear Dr. Nyholm,

Thank you for submitting your manuscript to PLOS ONE. After careful consideration, we feel that it has merit but does not fully meet PLOS ONE’s publication criteria as it currently stands. Therefore, we invite you to submit a revised version of the manuscript that addresses the points raised during the review process. Two expert reviewers in the field have evaluated your manuscript. I want to encourage you to adress the minor points raised by them.

We look forward to receiving your revised manuscript.

Kind regards,

Andreas Schäfer

Academic Editor

PLOS ONE

Journal Requirements:

Reviewers' comments:

Reviewer's Responses to Questions

**Comments to the Author**

1. Is the manuscript technically sound, and do the data support the conclusions?

Reviewer #1: Yes

Reviewer #2: Yes

2. Has the statistical analysis been performed appropriately and rigorously? 

Reviewer #1: Yes

Reviewer #2: Yes

3. Have the authors made all data underlying the findings in their manuscript fully available?

Reviewer #1: No

Reviewer #2: Yes

4. Is the manuscript presented in an intelligible fashion and written in standard English?

Reviewer #1: Yes

Reviewer #2: Yes

5. Review Comments to the Author

Reviewer #1: I congratulate the authors for the submitted work in which the evaluation of the pupils is examined in comparison of the conventional manual measurement to the apparative measurement. Overall, I think it is a promising article. However, I have a few small comments.

1. i would suggest to choose a concise title that shows the focus of this work (comparison with classical manual measurement to quantitative pupil measurement).

2. page 4 line 44-48: Please change the sentence structure. Discontinuation of therapy and good or poor neurological outcome are no synonymous. Poor outcome does not lead automatically to discontinuation of therapy (think about CPC class 3 and 4 or Mod Rank score of 3-5)

3. in page 6, lines 71-72: is the patient population studied OHCA patients, or does it also include electively intubated ventilated patients in the context of cardiogenic shock?

4. page 7 line 88: Please explain PLR

Page 8 Line 119: How many patients had withdrawal of life-sustaining treatment? Did the results of pupillometry lead to an extension of the diagnosis? Were there differences between classical and apparative measurement here?

5: Page 11, line 168-72: Very misleading! Is the mean age of the 8 male patients reflected?

You write that 71% resuscitated from OHCA. 71% of 14 patients would result in 9.94 patients. This does not make sense.

To what degree were the patients cooled? At what time (patient core body temperature) were the measurements taken?

6. In the tables, the values should be given as Mean ±SD and the percentages in parentheses. Otherwise seemed very confusing.

7. In: page 12 Table 1: Male sex with 8 of 14 corresponding to 57% and not 62%, and 7 of 14 equal to 50% for percutaneous coronary intervention (missing the word intervention here).

8. i have a question about the admission diagnosis in table 1. were all patients now cardiopulmonary resuscitated? It was stated that all patients had out-of-hospital cardiac arrest. According to this, how should cardiogenic shock or ST-segment elevation myocardial infarction and epidural hematoma be interpreted? (Does it make sense to compare a patient with epidural hematoma with patients with diagnoses from the cardiology field?)

9. page 12 line 174: The heading of the section should start on a new page. Again, the question arises at what time and under what circumstances the measurements were taken (Immediately after arrival?, Under hypothermia at 36 degrees? 32 degrees?).

10. page 14 Table 2: Again, values should be expressed as mean plus minus SD.

11. since 19 line 256 ... primarily with cardiac OHCA... Did all patients have OHCA or not? If necessary, only specialize on one group here to avoid bias.

12. page 22 Limitations. This section is too long and partly redundant. For example, the content of page 21 lines 310-316 is already in the Discussion section.

Some grammatical reworking is necessary, e.g.

in the title: repeatability, and reliability. The comma before the and should be removed if this heading is to be retained. Page

2, line 18: repeatability, and... as well as: Background: these when sitting should not be linked to a but. -Page

5 line 64: repeatability, ... (See above)-Page

8 line 115: Remove the colons from the heading-Page

8 line 116:... were,selected... (The comma must go)

-Page 17 line 239: Again:repeatability, and ... (The comma character !)

Reviewer #2: The authors studied validity of manual and automated pupillometry in 14 sedated and comatose patients and found that manual evaluation was less accurate in small pupils and poorly performed in the assessment of reactivity in 1 out of 4 evaluations.

Although limited by small sample size and the selected cohort, the study is of interest and useful for the broadened use of pupillometry in critical care patients

I have only minor comments to the authors:

Please provide data on absolute pupil size.

Was there a difference in reactivity among pts with and without opioid treatment?

Any difference among NPI>3 and < 3 for the comparison with manual assessment?

6. PLOS authors have the option to publish the peer review history of their article (what does this mean?). If published, this will include your full peer review and any attached files.

Reviewer #1: No

Reviewer #2: No

---

## [Author Response · Author response to Decision Letter 0]

5 Apr 2022

Dear editor and reviewer,

Following your letter regarding our manuscript “Observer and device reproducibility, repeatability, and reliability of automated quantitative pupillometry in critically ill cardiac patients”, we are sending this response letter explaining the changes made in the revised manuscript. 

First, we would like to thank you for taking your valuable time to evaluate our manuscript and for the insightful comments. Below is given a point-by-point response to the comments of the reviewers. 

All page and lines references refer to the manuscript before edited.

Academic Editor: When submitting your revision, we need you to address these additional requirements.

Response from authors:

We have applied these style requirements as requested.

Response from authors:

There should be no citation for retracted papers, hence, no changes has been made.

Response from authors:

We have updated our Data Availability statement in the editorial manager accordingly.

Response from authors

Our ethics statement resides in the sub-section “Patient assessment procedure” in the “Methods” section. 

Reviewer #1: I congratulate the authors for the submitted work in which the evaluation of the pupils is examined in comparison of the conventional manual measurement to the apparative measurement. Overall, I think it is a promising article. However, I have a few small comments.

1. I would suggest to choose a concise title that shows the focus of this work (comparison with classical manual measurement to quantitative pupil measurement).

Response from authors:

The suggestion is much appreciated, and we recognize that the title must be changed to better emphasize the subject of the manuscript.

Changes in the revised manuscript:

We have changed the full title (title page) to “Superior reproducibility and repeatability in automated quantitative pupillometry compared to standard manual assessment, and quantitative pupillary response parameters present high reliability in critically ill cardiac patients”, and the short title from “Observer and device reproducibility, repeatability, and reliability of automated quantitative pupillometry” to “Superior reproducibility and repeatability in automated quantitative pupillometry compared to manual”.

2. page 4 line 44-48: Please change the sentence structure. Discontinuation of therapy and good or poor neurological outcome are no synonymous. Poor outcome does not lead automatically to discontinuation of therapy (think about CPC class 3 and 4 or Mod Rank score of 3-5)

Response from authors:

We thank the reviewer for this valid point and will rephrase the mentioned sentence.

Changes in the revised manuscript:

The sentence at page 4 line 42-48 has been changed to the following: “It is challenging to identify these patients with poor neurological outcome early [7,8], and evaluation of pupil size and reactivity is of great prognostic importance [5,6]. When additional neurological evaluation (imaging of the brain, electroencephalography and somatosensory evoked potential) is needed before deciding about withdrawal of life-sustaining therapy, the timing can be guided by results from serial pupillometry as part of accurate multidisciplinary neuroprognostication [5,6,9]. Hence, great reliability for pupillometry is essential.”

3. in page 6, lines 71-72: is the patient population studied OHCA patients, or does it also include electively intubated ventilated patients in the context of cardiogenic shock?

Response from authors:

We thank the reviewer for this relevant question. This issue has further been raised in question #8 and #11 as well. To collectively clarify this, we have addressed all questions within the answer for question #11.

4. page 7 line 88: Please explain PLR

Page 8 Line 119: How many patients had withdrawal of life-sustaining treatment? Did the results of pupillometry lead to an extension of the diagnosis? Were there differences between classical and apparative measurement here?

Response from authors:

Regarding the first issue we thank you for pointing out this mistake which will be corrected. Concerning the second issue, of the 14 patients included 5 patients died (4 due to anoxic brain injury, and 1 due to refractory cardiogenic shock) at the ICU and all had withdrawal of life-sustaining treatment (WLST) prior to this. In the cases of anoxic brain injury, the decision WLST did rely upon a multidisciplinary assessment of neuron specific enolase, imaging of the brain, electroencephalography and somatosensory evoked potential combined thorough neurological examination. The results of pupillometry did not lead to an extension of the diagnosis. 

Changes in the revised manuscript:

The “PLR” at page 7 line 88 has been replaced with “pupillary light reflex (PLR)”.

5: Page 11, line 168-72: Very misleading! Is the mean age of the 8 male patients reflected?

You write that 71% resuscitated from OHCA. 71% of 14 patients would result in 9.94 patients. This does not make sense.

To what degree were the patients cooled? At what time (patient core body temperature) were the measurements taken?

Response from authors:

We thank the reviewer for making us aware of this need for clarification. We understand that the phrasing of the baseline characteristics leads the reader to believe that the statements refers only to the 8 males, instead of the total of the population.

The issue of TTM and timing of measurement have been raised again in question #9 where we have elaborated on this matter.

Changes in the revised manuscript:

We have rephrased the sentence at page 11 lines 168-172 to the following: “We included 14 patients, either sedated or comatose, with a mean age of 70±12 years. Patients were predominantly males (57%), 10 (71%) were resuscitated from OHCA, and 4 were other critically ill patients admitted to the cICU. We collected 56 sets of quadruple measurement-pairs for analysis, wherein 14 (25%) pairs were obtained while the patients underwent TTM to 36 degrees Celsius, 44 (79%) while patients were sedated, and 46 (82%) while patients received a vasoactive agent. All baseline characteristics are presented in Table 1.”

6. In the tables, the values should be given as Mean ±SD and the percentages in parentheses. Otherwise seemed very confusing.

Response from authors:

We agree that this setup can be confusing and thank the reviewers for emphasizing this point. 

Changes in the revised manuscript:

In Table 1 (page 12) we have replaced alle parentheses containing mean values with “value±SD” instead and left the percentages in parentheses. 

7. In: page 12 Table 1: Male sex with 8 of 14 corresponding to 57% and not 62%, and 7 of 14 equal to 50% for percutaneous coronary intervention (missing the word intervention here).

Response from authors:

These are of course miscalculations and a neglection of a missing word on our part. We thank you for pointing out these errors.

Changes in the revised manuscript:

In page 12 Table 1, “Male sex” we have replaced the percental value from “62” to “57”, replaced “Percutaneous coronary” with “Percutaneous coronary intervention” and the percental value from “47” to “50”,

8. I have a question about the admission diagnosis in table 1. were all patients now cardiopulmonary resuscitated? It was stated that all patients had out-of-hospital cardiac arrest. According to this, how should cardiogenic shock or ST-segment elevation myocardial infarction and epidural hematoma be interpreted? (Does it make sense to compare a patient with epidural hematoma with patients with diagnoses from the cardiology field?)

Response from authors:

Again, we thank the reviewer for this question and refer to the answer for question #11, later in the text. 

9. page 12 line 174: The heading of the section should start on a new page. Again, the question arises at what time and under what circumstances the measurements were taken (Immediately after arrival?, Under hypothermia at 36 degrees? 32 degrees?).

Response from authors:

We thank the reviewer for emphasizing these points. We agree that the heading should be moved to a new page.

Regarding the circumstances the measurements in #5 and in #9 we have added the temperature of our TTM regime in rewriting the first sentence of the result section. The timing of the individual measurements is discussed in the “Methods” section of the manuscript. They were obtained “in the period from admission through the initiation and discontinuation of sedation, targeted temperature management (TTM), and treatment with vasoactive agents, to discharge from the cICU or withdrawal of life-sustaining treatment.” (page 8 line 117-119).

For OHCA patients treated with TTM, some measurements were obtained during TTM and some after TTM as well. For non-OHCA no measurements were obtained during TTM. In all, 14 (25%) of all measurements in this study were obtained during TTM (36 degree Celsius), and 48 (75%) were obtained from a patient not treated with TTM at that moment.

Changes in the revised manuscript:

At page 12, line 174 the heading “Standard manual versus quantitative pupillometry - Pupil size”, have been moved to page 13, line 175. We have added “to 36 degrees Celsius” at page 8 line 118.

10. page 14 Table 2: Again, values should be expressed as mean plus minus SD.

Response from authors:

Again, we thank the reviewer pointing out this issue.

Changes in the revised manuscript:

As with Table 1, we In Table 2 (page 14) we have also replaced alle parentheses containing mean values with “value±SD” instead and left the percentages in parentheses. 

11. since 19 line 256 ... primarily with cardiac OHCA... Did all patients have OHCA or not? If necessary, only specialize on one group here to avoid bias.

Response from authors:

In the context of question #3, #8 and #11, we acknowledge that the inclusion criteria are not presented clearly regarding patient population and thank the reviewer for emphasizing this. 

Question #3 In this study we included all patients admitted to our cardiac ICU, primarily OHCA patients but also patients with cardiogenic shock and other hemodynamically instable patients requiring specialized intensive care. No elective patients were included. 

Question #8: Regarding the admission diagnosis in table 1, we admitted 10 patients resuscitated from OHCA, 2 patients with ongoing cardiogenic shock, 1 hemodynamically unstable patient with STEMI, and 1 hemodynamically unstable patient initially admitted with epidural hematoma. Hence, not all patients were OHCA patients included were OHCA patients, but the vast majority were.

Answer for question #11: As the results state, we included 10 OHCA patients and 4 “other” patients according to patients admitted at the cardiac ICU in the study period. 

The overall concern seems to be whether it makes sense to compare patients with different diagnosis (#8) and how to avoid subsequent bias if doing so (#11). This is a very valid point and we thank the reviewer for bringing this up.

In this study we focus on validating the measuring methods of standard manual and quantitative pupillometry in the clinical setting. Whether it yields reproducible data between measurements, observers and devices or not. Our setup does not examine pupillometry in regards of patient outcome.

The concern should then be if the pupillometry would yield the same reproducibility, repeatability, and reliability for OHCA than for different diagnosis and these consequently diluting the signal from the primary group. However, several earlier studies have investigated the intra/inter- and inter-device reliability separately for different diagnosis yielding the same high reliability for quantitative pupillometry and superiority compared to manual investigation (Phillips et. al, Neurocrit Care, 2019). Even studies using healthy controls (Couret et al., Critical Care, 2016). Thus, we had less concern for this bias in our validation study. Nevertheless, we have compared the reliability of quantitively assessed size for both intra-, and interobserver measurements for OHCA and non-OHCA patients. 

With intraobserver measurements the OHCA group yielded an ICC (95%CI) at 0.99 (0.98-0.99) and non-OHCA group at 0.97 (0.86-0.99), with no significant difference between means of the two groups (p=643). With interobserver measurements we found ICC at 0.97 (0.96-0.98) for the OHCA group, and 0.94 (0.72-0.99) non-OHCA group. Again, with no significant difference between means of the two groups (p=0.324).

Changes in the revised manuscript:

We have changed the sentence at page 6, lines 71-72 to the following: “Throughout August and September 2020, all patients admitted to the cICU were considered eligible for inclusion and otherwise treated in agreement with guidelines [7]. This comprised hemodynamically unstable patients requiring specialized intensive cardiac care. No elective patients were included.”

12. page 22 Limitations. This section is too long and partly redundant. For example, the content of page 21 lines 310-316 is already in the Discussion section.

Response from authors:

We have looked through page 22 “Limitations” and have some difficulty finding the redundance in this section. If the reviewer feels that the redundance is within this section, we much ask for a specific reference. 

However, the reviewer makes a point that the content of page 21 lines 310-316 is already in the “Discussion”, hence we interpret the inquiry to be with this section. We thank the reviewer for making this point aware to us and will revise the section accordingly.

Changes in the revised manuscript:

We have rewritten the Discussion section with the following editions

In page 21 line 310-316 we have corrected the sentence the following: “For all the additional quantitative pupillary response parameters, recently presented with promising prognostic value by Tamura et al. [19], this study offers novel data on reproducibility and repeatability.”

In page 22 line 318-324 we have corrected the sentence to the following: “Overall, this study support that automated quantitative pupillometry performs well in the clinical setting even when several observers, multiple patients and separate pupillometry devices are involved. However, further studies, implementing the additional quantitative pupillary response parameters, are needed.”

Some grammatical reworking is necessary, e.g.

in the title: repeatability, and reliability. The comma before the and should be removed if this heading is to be retained. 

-Page 2, line 18: repeatability, and... as well as: Background: these when sitting should not be linked to a but. 

-Page 5 line 64: repeatability, ... (See above)

-Page 8 line 115: Remove the colons from the heading

-Page 8 line 116:... were,selected... (The comma must go)

-Page 17 line 239: Again:repeatability, and ... (The comma character !)

Response from authors:

We recognize these grammatical errors and thank the reviewer for pointing this out.

Changes in the revised manuscript:

We have made the following corrections in the manuscript:

The title has been changed in answer for question #1.

At page 2 line 17-19 we have rewritten the paragraph: “Quantitative pupillometry is part of multimodal neuroprognostication of comatose patients after out-of-hospital cardiac arrest (OHCA). However, the reproducibility, repeatability, and reliability of quantitative pupillometry in this setting have not been investigated.” 

We removed a “,” at page 5 line 64, page 8 line 116, page 17 line 239, and removed a “:” at page 8 line 115. 

Reviewer #2: The authors studied validity of manual and automated pupillometry in 14 sedated and comatose patients and found that manual evaluation was less accurate in small pupils and poorly performed in the assessment of reactivity in 1 out of 4 evaluations.

Although limited by small sample size and the selected cohort, the study is of interest and useful for the broadened use of pupillometry in critical care patients. I have only minor comments to the authors:

Please provide data on absolute pupil size. 

Response from authors:

In this study we refer to the absolute pupil size as the pupillometer parameter “maximum pupil size (MAX)”, according to “NPI®-200 Pupillometer System - Instructions for Use” by NeurOptics®. Mean values of absolute pupil size is provided in “Table 2” under the subheading “Maximum diameter, mm” for both “Manuel Pupillometry” and “Quantitative Pupillometry”. However, we fully acknowledge that the term “maximum pupil size” can be ambiguous and will make an adjustment to clarify this. We thank the reviewer for emphasizing this.

Changes in the revised manuscript:

We have added “/absolute” to the sentence in page 7 line 91: “…velocity (DV, millimeters/seconds), besides maximum/absolute and minimum pupil size (MAX/MIN, millimeters) [22].”

Was there a difference in reactivity among pts with and without opioid treatment?

Response from authors:

This is very interesting and relevant point, and we thank the reviewer for bringing this up. 

We would expect that the mean value of reactivity (quantitively measured as %CH) were lower in the sedated patients. However, we made analyses to investigate if pupillometry were still reliable under sedation. 

Overall we found a difference in mean values between sedated and unsedated patients, however the difference were not significant, and although the interobserver reliability were slightly lower for the sedated patients, all measurements had excellent reliability, in regards of the usual criteria (Portney &Watkins, 2000). 

Changes in the revised manuscript:

We have added “In measurements performed while patients were sedated the mean %CH value was 17% and 25% when not sedated, although lower with sedation, this difference was not significant (p= 0.234). When testing reliability for measurements without sedation, we found an ICC at 0.98 (0.95-0.99) and 0.99 (0.96-0.99) for intra-observer and inter-observer measurements, respectively. For the sedated we found ICC at 0.96 (0.94-0.98) and 0.92 (0.86-0.95) for intra-observer and inter-observer measurements.” to page 16 line 222 and “even under sedation,” to page 20 line 291.

Any difference among NPI>3 and < 3 for the comparison with manual assessment? 

Response from authors:

Again, the reviewer raises a very interesting focus. However, in this study the lowest NPi value measured were 3.3, hence we obviously could not calculate any difference below. We ran test for NPI>4 and <4 regarding reliability of measurements between observers for pupil size.

For quantitative pupillometry mean values of pupil size were 2.23mm for NPi > 4, and 2.00mm for NPi < 4 (p-value = 0.029). For manual assessed pupil size, mean pupil size were 2.29mm for NPi > 4, and 1.87mm for NPi < 4 (p-value < 0.001). When testing reliability for quantitative pupillometry for NPi > 4 we found an ICC at 0.99 (0.99-1.00) and 0.99 (0.98-0.99) for intraobserver and interobserver measurements, respectively. For NPi < 4 we found ICC at 0.97 (0.91-0.99) and 0.97 (0.89-0.99) for intraobserver and interobserver measurements, respectively. The manual assessment yielded for NPi > 4 yielded ICC at 0.94 (0.90-0.97) and 0.91 (0.85-0.95) for intraobserver and interobserver measurements, respectively. For NPi < 4 we found ICC at 0.78 (0.28-0.9) and 0.80 (0.45-0.94) for intraobserver and interobserver measurements, respectively. 

Overall, we observed worse ICC for measurements with NPi < 4, especially for manual 

Pupillometry. This important tendency should be further investigated in a larger population, in future studies, for NPi value below and above 3.

---

## [Decision Letter · Decision Letter 1]

18 Jul 2022

Superior reproducibility and repeatability in automated quantitative pupillometry compared to standard manual assessment, and quantitative pupillary response parameters present high reliability in critically ill cardiac patients

PONE-D-22-00255R1

Dear Dr. Nyholm,

We’re pleased to inform you that your manuscript has been judged scientifically suitable for publication and will be formally accepted for publication once it meets all outstanding technical requirements.

Kind regards,

Andreas Schäfer

Academic Editor

PLOS ONE

Additional Editor Comments (optional):

Reviewers' comments:

Reviewer's Responses to Questions

**Comments to the Author**

1. If the authors have adequately addressed your comments raised in a previous round of review and you feel that this manuscript is now acceptable for publication, you may indicate that here to bypass the “Comments to the Author” section, enter your conflict of interest statement in the “Confidential to Editor” section, and submit your "Accept" recommendation.

Reviewer #1: All comments have been addressed

Reviewer #2: All comments have been addressed

2. Is the manuscript technically sound, and do the data support the conclusions?

Reviewer #1: Yes

Reviewer #2: Yes

3. Has the statistical analysis been performed appropriately and rigorously? 

Reviewer #1: Yes

Reviewer #2: Yes

4. Have the authors made all data underlying the findings in their manuscript fully available?

Reviewer #1: Yes

Reviewer #2: Yes

5. Is the manuscript presented in an intelligible fashion and written in standard English?

Reviewer #1: Yes

Reviewer #2: Yes

6. Review Comments to the Author

Reviewer #1: I thank the authors for this revised work. All objections have been satisfactorily dealt with. I have no new objections

Reviewer #2: All points have been adequately adressed in the revised version of the manuscript. I have no further comments

7. PLOS authors have the option to publish the peer review history of their article (what does this mean?). If published, this will include your full peer review and any attached files.

Reviewer #1: **Yes: **Dr. Muharrem Akin

Reviewer #2: No

---

## [Editor Report · Acceptance letter]

20 Jul 2022

PONE-D-22-00255R1 

Superior reproducibility and repeatability in automated quantitative pupillometry compared to standard manual assessment, and quantitative pupillary response parameters present high reliability in critically ill cardiac patients 

Dear Dr. Nyholm:

I'm pleased to inform you that your manuscript has been deemed suitable for publication in PLOS ONE. Congratulations! Your manuscript is now with our production department. 

Kind regards, 

on behalf of

Prof. Dr. Andreas Schäfer 

Academic Editor

PLOS ONE